# Expert Finding Systems: A Systematic Review

**Omayma Husain** [1,2,*], **Naomie Salim** [1], **Rose Alinda Alias** [3] 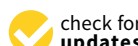, **Samah Abdelsalam** [1,2] and **Alzubair Hassan** [4]

1 School of Computing, Faculty of Engineering, Universiti Teknologi Malaysia, Skudai 81310, Malaysia; naomie@utm.my (N.S.); sama7_279@yahoo.com (S.A.)

2 Department of Computer Science, University of Khartoum, Khartoum 11111, Sudan

3 Azman Hashim International Business School, Universiti Teknologi Malaysia, Skudai 81310, Malaysia; alinda@utm.my

4 School of Computer Science and Cyber Engineering, Guangzhou University, Guangzhou 510006, China; alzubairuofk@gmail.com

\* Correspondence: omaymahusain@gmail.com or hahomayma@live.utm.my

**Abstract:** The data overload problem and the specific nature of the experts' knowledge can hinder many users from finding experts with the expertise they required. There are several expert finding systems, which aim to solve the data overload problem and often recommend experts who can fulfil the users' information needs. This study conducted a Systematic Literature Review on the state-of-the-art expert finding systems and expertise seeking studies published between 2010 and 2019. We used a systematic process to select ninety-six articles, consisting of 57 journals, 34 conference proceedings, three book chapters, and one thesis. This study analyses the domains of expert finding systems, expertise sources, methods, and datasets. It also discusses the differences between expertise retrieval and seeking. Moreover, it identifies the contextual factors that have been combined into expert finding systems. Finally, it identifies five gaps in expert finding systems for future research. This review indicated that ≈65% of expert finding systems are used in the academic domain. This review forms a basis for future expert finding systems research.

**Keywords:** expert finding systems; expertise retrieval; expertise seeking

## 1. Introduction

In the past few years, several organisations have realized that effective management of all data assets could help them survive in the competitive business environment. Expertise is an important knowledge asset that is often stored in people's mind and is therefore difficult to codify. Expertise can be shared when people interconnect with one another. Users need to consult an expert to determine ways to solve their problems in a particular domain, such as medical problems. Although there is a huge volume of data available for solving problems, people still seek the services and guidance of an expert. For example, conference planners search for paper reviewers, and students require appropriate supervisors for their research [1].

The term 'expertise' is defined in psychology as the human cognitive skills that can be developed by conducting a task frequently [2]. Generally, people who display expertise in a specific field are known as experts. However, there is no unified definition of the term 'expert'. The authors in reference [3] defined an expert as an individual who has skills in his domain of expertise and can evaluate a particular domain topic. Furthermore, the authors in reference [4] identified an expert as an individual who has a great deal of knowledge and displays comprehensive skills in a specific area. The authors in reference [5] identified an expert as a person who has the best experience about a particular topic in a particular field of study, based on a collection of various factors.

The expertise seekers are those who seek knowledge for specific purposes such as problem-solving, collaboration, and team formation. They require comprehensive information regarding experts who can answer their questions. It is not practical to manually collect information regarding experts, specifically in widely distributed and large-scale organisations [6,7]. Therefore, Information Retrieval (IR) techniques have been used to facilitate the retrieval of the appropriate experts automatically. Such techniques are called expert locating systems (also known as expert finding systems or expertise retrieval). Expert finding systems are IR systems that identify candidate experts and ranking them based on their expertise in a given subject. The candidate's expertise is extracted from their expertise evidences such as publications, reports, projects, and social network [8,9].

The expertise retrieval system is useful for individuals and organisations, as it allows them to retrieve suitable experts. Hence, these systems help the expertise seeker to find the most appropriate expert, based on the expert's actual knowledge [10]. Early expert finding systems developed to focus on specific document types. For instance, the authors in references [11,12] studied locating experts in the technical and support departments of a software corporation. The authors in reference [13] developed a tool called Expertise Browser for locating expertise in a collaborative software engineering environment. Because of clear limitations in the above systems (i.e., concentrating on specific document types), academia and industry have given greater attention to systems that have capabilities to mine heterogeneous sources of expertise evidence accessible in an organization. The P@noptic system proposed by the authors in reference [14] is one of the first approaches of this kind of system. The P@noptic system confirmed the feasibility of expertise retrieval on heterogeneous collections. As a result, at the Text Retrieval Conference (TREC) from 2005 to 2008, an expert finding task was launched as a portion of the Enterprise Track. It provided expert finding systems researchers with a public platform to evaluate approaches and algorithms developed for expert finding. Consequently, expert finding received significant attention from the IR research community, and quick progress was made in modeling, algorithms design, and evaluation.

Expert finding systems have been used in different domains and environments such as enterprise [9,15], question answering communities [16,17], an online knowledge community [18,19], social networks [20–24], and academia [25–28]. Experts finding systems present the following advantages [10]:

1. Individual Benefits—these systems, along with finding people, also link people who would otherwise not have the opportunity to meet in person. They could also increase the individual's awareness regarding the activities in the organisation and allow people to tailor their responses based on people they may meet in future.

2. Benefits for an organisation (use of conversation for retrieving tacit knowledge)—Expert finding systems help organizations in two ways: (1) They map tacit knowledge and help individuals to develop a better awareness about which individual knows what; however, information of the knowledge location alone is insufficient. The organisation benefits if this knowledge can be applied to actual problems. (2) Any expert finding system must help in building social capital by improving the connections between people who are acquainted with one another and increasing conversations between people who are not acquainted, thus extending their social network (i.e., development of weak or strong ties).

The expert finding task usually has procedures that can be classified into the following five issues:

1. Expertise evidence (sources of data) selection—Expertise evidence selection extracts the expertise-related data and information from which experts can be discovered. Data related to the person should be obtained to discover whether or not the individual is an expert in a particular field.

2. Expert representation—a key objective of expert finding systems is to provide users with information that effectively enables not only locating expertise but also deciding and selecting among relevant experts. It is necessary to identify which information can be useful in determining

among experts whose expertise seeker is not personally familiar with them. Generally, an expert can be determined based on his documented evidence and relationships and human interaction factors (which are called contextual factors).

3.  Model building—Model building consists of pre-processing, indexing, and modelling.

    a.  Pre-processing and indexing—constructing indexes is a regular task for document retrieval systems, and numerous open source software tools exist to facilitate this task; for expertise retrieval, however, there is a further challenge. The identifiers of a candidate expert (e.g., names, emails, etc.) need to be recognized in documents, and heterogeneous data sources need to be integrated to identify experts. Numerous text pre-processing techniques for traditional IR applications are implemented in expert search. For instance, several expert search systems eliminate stopwords from raw text document data, to achieve a higher level of precision [1]. An expert's data is commonly collected from many documents in heterogeneous data sources. This is a significant difference with document retrieval that produces technical challenges to both data integration and indexing. As an example, different names may refer to the same person or the same names in different sources might refer to different persons. Named entity recognition (NER) and disambiguation play a fundamental role in most operational expert search systems [1]. A simple way of carrying out NER is to use the candidate's identifiers as the query and associate the retrieved documents with that person [29]. Reference [30] proposed rule-based matching techniques to identify candidates by their names (using three matching forms—exact match, initial and last name, and last name only) and by their emails.

    b.  Modelling and retrieval—Modelling and retrieval are about building models to associate candidate experts with a user query and to provide a ranking based on the strength of these associations [1]. In the literature, there are several different methods for building models to find experts, such as probabilistic, network-based (or sometimes called graph-based), voting, and other models [1,4].

4.  Models evaluation—the expert finding system is an IR system where interactions start with a user sending a query to the system, and then the system retrieves a list of experts that hopefully are relevant to the user's query. Normally, expert finding system efficiency is evaluated using a test collection (datasets).

5.  Interaction design—the presentation of expert search results to users is a primary issue in practice. However, a simple list of names does not every time aid the user to judge the relevance of a candidate to the query. Contrary to document search, there is often no particular document snippet that can be speedily examined to determine relevance. For that reason, expert search result pages often show not only a ranked list of people but also a list of documents, conferences, and journals. Also, the presence of contact details for each ranked expert is important to facilitate communication [1]. There is a significant number of studies on locating personal homepages where several of the suggested methods utilize supervised learning techniques [31–33]. Also, photos of experts are essential because users may need to determine the likely seniority or familiarity of an expert in advance of contacting him or her. For instance, the user may be looking for somebody of a similar age or level or type of experience to themselves. Moreover, related information such as affiliations and documents, containing publications, and projects, appear to help the user determine that the expert is likely to have relevant expertise [1].

Usually, there are two important expert finding systems tasks— (1) expert finding concentrating on identifying a particular expert in a specific domain and (2) expert profiling identifying the area of expertise of the individual experts. Earlier studies [1,4] stated that these tasks are two sides of the same coin, since, in combination, they lead to a ranking problem, wherein the collection of the objects that had to be ranked were prioritized.

Despite these advances and achievements in expert finding systems research, we discovered only one survey study about expert finding systems published in 2017 [4] that analyzed expert finding systems methods and introduced data collections (datasets) used for evaluating expert finding systems. However, only six studies between 2010 and 2013 were included in the survey. Our study presents a systematic review, which includes 96 studies between 2010 and 2019. It focuses on answering five research questions represented in Section 5. In the first question, it tries to find out common domains of expert finding systems. The second question focuses on expertise sources that have been used to identify experts. The third question concentrates on the difference between expertise retrieval and seeking and theories and contextual factors that are combined into expert finding systems to improve their effectiveness. The fourth question focuses on the state-of-the-art methods that have been used for models building. Finally, the fifth question concentrates on the datasets that have been used to evaluate expert finding systems. We found gaps in efforts to identify and analyse the contextual factors and to identify the difference between expertise retrieval and seeking, as well as to identify the domains that used expert finding systems. Moreover, new methods and datasets since 2013 were not included. Hence, there is a need to generate evidence that highlights the benefits of the existing studies. This review will be very helpful for future researchers and practitioners in expert finding systems and expertise seeking domains. The rest of this review is structured in the following manner: Section 2 describes the research method, Section 3 describes the results of the research questions, Sections 4 and 5 present research findings on identified gaps and threats to validity, and Section 6 presents the conclusions of the study.

## 2. Research Method

The researchers carried out this SLR based on the guidelines described by the authors in reference [34]. The research methodology flowchart is shown in Figure 1. There were four phases in this SLR paper. The first phase concentrated mainly on brainstorming and identifying the research questions. Five research questions were identified from this phase. The second phase executed the search strategy, which combined two search techniques—automated and manual search protocols. The third phase selected the papers based on analysing a collection of potential papers. The fourth phase then performed data synthesis and answered the predefined questions.

### 2.1. Research Questions

The main objectives of this study were to study, understand, and summarize current studies on expert finding systems to identify the domains that applied these systems, the expertise sources used, the methods and data sets, the differences between expertise retrieval and expertise seeking, and the contextual factors and theories integrated to expert finding systems. Additionally, it critically identifies the possible research gaps for future studies in this area. In order to accomplish these objectives, this study identified five research questions (RQ) relating to the aim of this study, listed in Table 1.

**Table 1.** Research questions.

| Code | Research Questions |
| --- | --- |
| RQ1 | Which domains use the expertise retrieval systems? |
| RQ2 | What are the expertise sources used for experts finding systems? |
| RQ3 | What are the differences between expertise retrieval and expertise seeking, and what are the contextual factors and theories that have been integrated with expert finding systems? |
| RQ4 | What are the methods that have been used for expert finding systems? |
| RQ5 | What are the data sets that have been used for expert finding systems? |

### 2.2. Search Strategies

In this study, the search included two search strategies, manual and automatic, as described in Figure 1. The researchers used an automated search process for determining the major studies,

which included several extra studies for acquiring a broader perspective. The authors of reference [34] recommended conducting a manual search on the major references. In the first stage, the researchers explored the following databases: Science Direct, Wiley, Springer Link, Scopus, ACM, IEEE Explore, Tylor, and the Web of Science. They selected these databases because they were believed to be the most relevant and consisted of journal articles and conference proceedings, with the highest impact factor. This could offer a better idea of the type of expertise seeking and expertise retrieval techniques used by different research groups. Based on the review structure and the research questions, the researchers used a few specific keywords for identifying the relevant articles, which were as follows: Expert recommendation systems OR Expertise retrieval OR Expert finding systems OR Expert allocation systems OR Expertise finding OR Expertise seeking OR Expertise selection OR Expert selection OR Expertise source selection OR Interpersonal source selection.

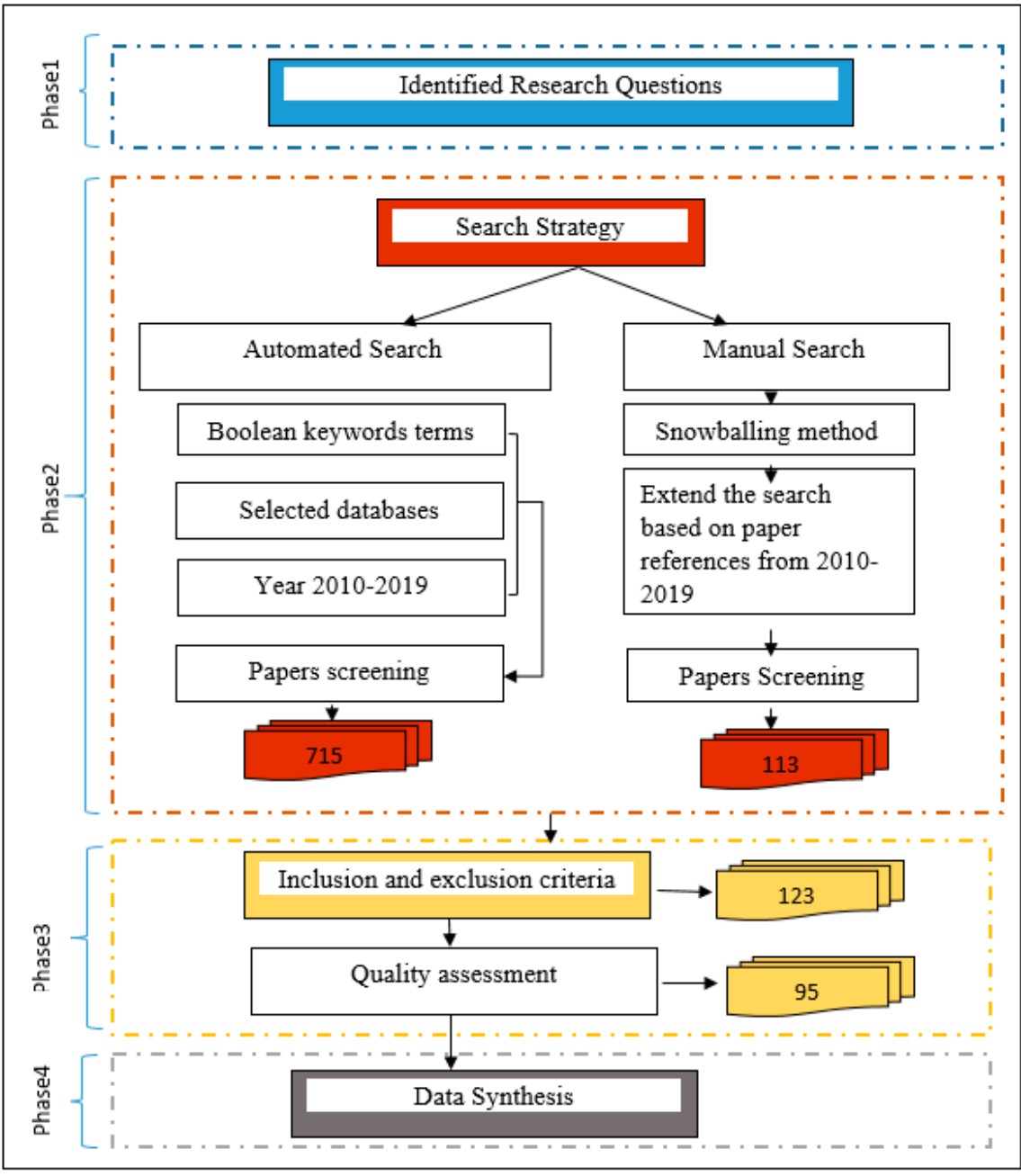

**Figure 1.** Research method.

After deriving the initial data from the above-mentioned databases, with the help of a search string, the researchers conducted a manual search. The manual searching strategy were performed through the 'snowballing method'. Both forward and backward snowballing searches were performed over references of identified papers from the same duration. They also used the Google Scholar search engine to identify studies that were previously cited in the primary studies. A manual search ensured that this SLR was complete and comprehensive. This also helped in eliminating duplication.

### 2.3. Study Selection Process

With the above-mentioned search engines using the specific keywords, the authors identified 716 studies using an automatic search process and 113 studies using the manual search process. Out of the total 828 studies, 199 studies were duplicates and hence were eliminated using EndNote. The remainder of the 630 studies were further analysed using the inclusion and exclusion criteria. The authors set up the inclusion and the exclusion criteria for ensuring that the system only included the papers that were relevant for the study. Table 2 summarizes all the inclusion and exclusion criteria. It must be noted that the study did not fulfil any of the exclusion criteria but satisfied all the inclusion criteria. The articles not included in any journals, workshop, or conference proceeding were eliminated. Here, seven PDF studies were unavailable, while 498 studies were discarded based on their titles and abstracts. This left 124 studies for quality assessment.

**Table 2.** Inclusion and exclusion criteria.

| Inclusion Criteria | Exclusion Criteria |
| --- | --- |
| Study manuscript written in English | Not in English |
| Published in years (2010–2019) and in the selected databases | Duplicated studies |
| Full-text | Articles that did not match the inclusion criteria |
| Directly or indirectly answers the defined research question(s) | Not related to the research questions of this study |

### 2.4. Quality Assessment (QA)

The QA helped in determining the general quality of all the selected studies [34]. Hence, for determining the reading strength and analysing the results of the search processes, the researchers used the following QA criteria for evaluating all the studies:

QA1: Are all the topics addressed in the research paper related to the review?
QA2: Does the research paper explain the context clearly?
QA3: Is the research methodology explained in the paper?
QA4: Does the paper describe the data collection method?
QA5: Does the paper present the result analysis?

The authors used these five QA criteria for assessing the 124 studies for determining the credibility of the selected studies. The quality of the paper was assessed by scoring each QA criteria as high, medium, or low (Figure 2). If the study satisfied a criterion, it was given a score of 2. However, if it partially satisfied the criterion, it was scored 1, while it was scored 0 if it did not satisfy the criterion. The paper quality was deemed to be high if it could score value of ≥6, while it was considered to be medium if it scored 5, and was considered to be of lower quality if it scored <5. During QA, 28 studies did not satisfy the criteria and hence were excluded from the final list of papers. Thus, the SLR consisted of 96 papers, wherein several studies scored a higher QA value as presented in Figure 2.

### 2.5. Data Extraction and Synthesis

The data extraction and data synthesis processes were carried out by reading all 96 papers and extracting the relevant data using EndNote and the MS Excel spreadsheets. In this stage, the researchers aimed to derive the data extraction forms for recording all the data acquired from the primary 96 papers. They established the following columns in the review: study ID for identifying every study, title,

author list, year of publication, publishing location (i.e., journals, conference proceedings, etc.), study context (for determining the topic areas which used the expert retrieval systems), methods, data sets, contextual factors used, and the various theories applied. All the factors were selected according to the objectives and the research questions. In this phase, the authors present important statistical results regarding the studies included in this review. The authors described the primary studies based on their sources, publication type (see Figure 3), and the research methodologies applied (see Figure 4).

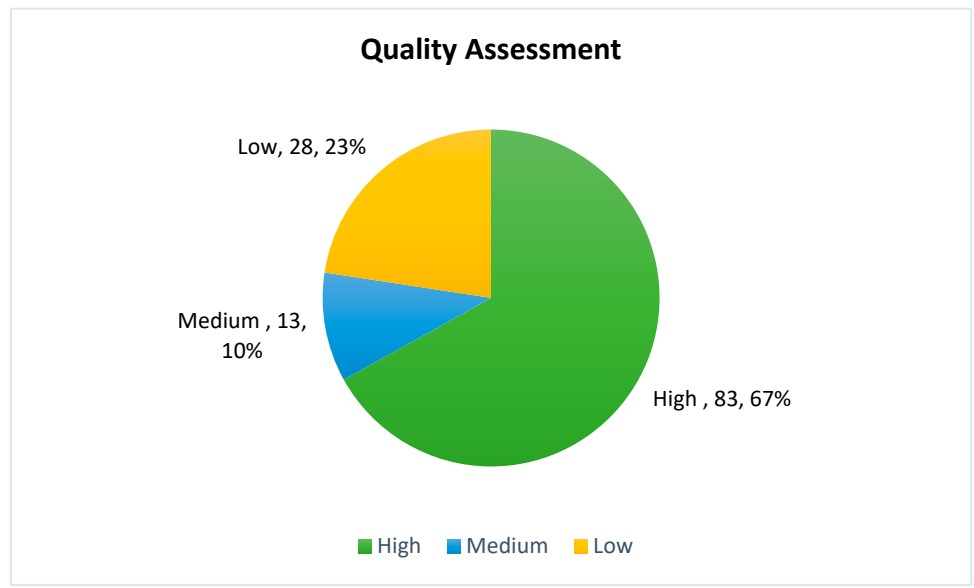

**Figure 2.** Quality Assessment.

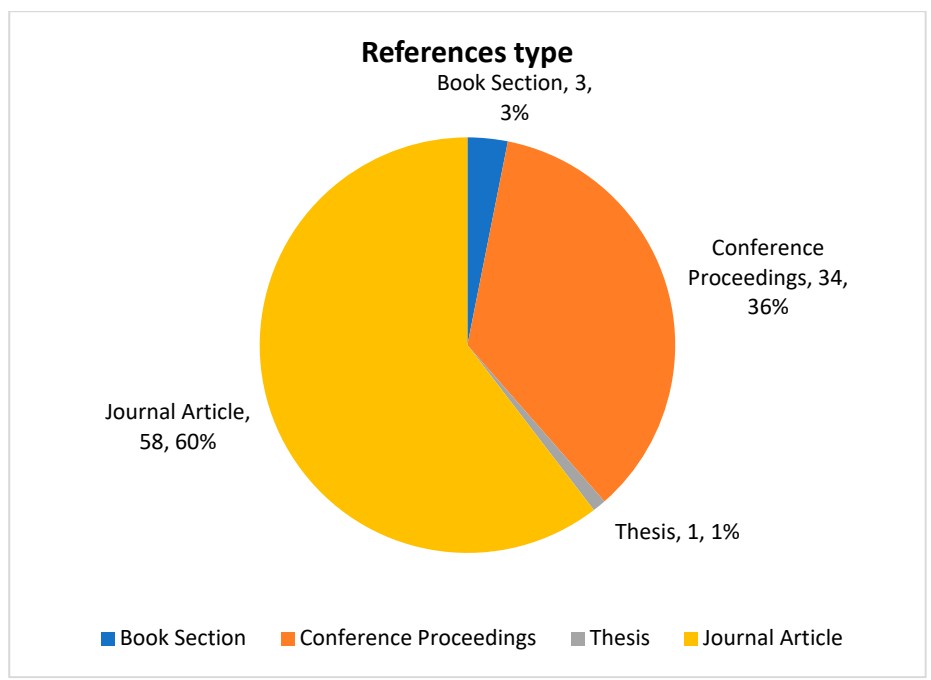

**Figure 3.** Type of references.

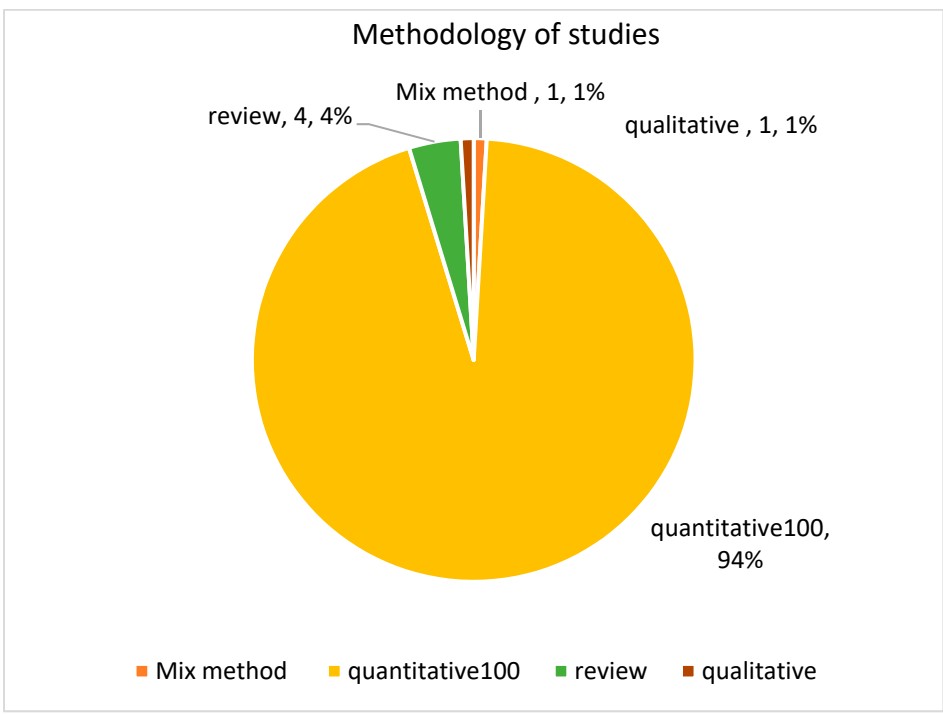

**Figure 4.** Research methodology.

### 2.5.1. Publications Source Overview

The majority of the primary studies were published and cited in journals and conferences. Figure 3 presents the distribution of these studies derived from their publication sources. It shows that the primary studies included 63 journal articles (58%), 40 conference reports (38%), three book chapters (3%), and one thesis (1%).

### 2.5.2. Research Methodologies

Figure 4 presents the distribution of all primary studies based on their research methodologies. It was seen that 90 articles included quantitative methods, i.e., (94%), one qualitative technique (1%), one mixed method (1%), and four reviews (4%).

## 3. Results for the Research Questions

After analysing and extracting the data from the primary studies, the authors proceeded to answer the research questions. All studies based on the research questions were mapped, and the similar studies were grouped. The following sub-sections provide the results for each research question.

### 3.1. RQ1: Which Domains Used the Expert Finding Systems?

The expert finding systems have been developed in different domains such as academia, enterprise, and social networks. For example, expert finding systems have been used in academia to find supervisors, paper reviewers, and research collaborators. In this SLR, the authors noted that 68 studies developed expert finding systems. After analysing all these studies, the researchers observed that these systems were developed in six different domains for different purposes. These domains are enterprises, academia, knowledge sharing communities (KSCs), medicine, social networks, and online forums. Table 3 describes these domains and studies in further details. From the table, we noted that academia is the largest domain, comprising 44 studies (65%). A majority of the expert finding systems were used in the academic domain. Additionally, in this field, the majority of the systems were developed for specific tasks like paper reviewing, research collaborations, finding similar experts, industry and

university collaborations, and finding a suitable supervisor. Enterprise is the next largest domain, consisting of 10 studies (15%). A majority of the studies in this domain applied expert finding systems for general tasks, except reference [35], wherein the researchers designed the expert finding system for offering formal help. In another study [36], an expert finding system was developed for developer recommendations. The third major domain is medicine, which consisted of two studies (3%). These studies focused on the development of expert finding systems for finding medical experts. The fourth domain is the online knowledge sharing communities (KSCs), consisting of six studies (9%). In the past few years, there has been a rapid development of the online KSCs, like question-answering (Q&A) communities, discussion boards, and blogs, which generate a massive amount of data and expertise. Online KSC users exchange ideas, share experiences, and use these communities for solving their issues. The fifth domain is online forums, consisting of two studies (3%). These studies developed expert finding systems for helping the users of such forums to find experts in a specific area. Social media is the sixth domain, with three studies (5%), which developed these systems for finding experts from various social networks like Twitter and Facebook.

As described in the academic domain, the expertise retrieval systems could be developed for a variety of tasks, which were described in 43 studies, presented in Table 3. Fourteen (30%) of the studies did not specify the tasks. Thirteen studies (30%) used the expert finding systems for research collaborations in the academic domain, followed by paper reviewing in nine studies (21%), finding group of experts in three studies (7%), university industry collaboration in two studies (5%), finding a similar expert in two studies (5%), and finding a supervisor in one study (2%).

**Table 3.** Expert finding systems' domains.

| The Domain | Task | Frequency | References |
|---|---|---|---|
| Academia University | Research collaboration | 1 | [37] |
| | Paper reviewing | 2 | [38,39] |
| | university-industry collaboration | 1 | [25] |
| | Supervisor finding | 1 | [26] |
| | Similar researcher | 2 | [27,40] |
| | No specific task | 6 | [41–46] |
| Academia Social Network | Research collaboration | 12 | [20,21,47–55] |
| | Paper reviewing | 7 | [5,21,56–60] |
| | Finding group of experts | 3 | [61–63] |
| | R&D | 2 | [64,65] |
| | No specific task | 8 | [66–73] |
| Enterprise | | 10 | [15,22,35,36,74–79] |
| Medicine | | 2 | [80,81] |
| Knowledge Sharing Communities | | 6 | [16,17,19,44,82,83] |
| Online forums | | 2 | [84,85] |
| Social network | | 4 | [56,57,86–88] |

### 3.2. RQ2: What Are the Expertise Sources Used for Experts Finding Systems?

First, in order to decide if a person is an expert in a particular area, we need to obtain relevant information about that person; this information is called expertise sources. Expertise sources can include publications, homepages, related documents, and social networks. In this study, expertise sources in current expert finding systems have been classified into the following three channels and are represented in detail in Table 4. The table contains the three channels, the expertise sources that relate to each channel, and the previous studies that used the expertise source.

1.  Textual sources—Textual sources can be considered as the main evidence of expertness. They include documents, user profiles (meta databases or homepages), and communications (e-mails and textual chats) [45]. Documents are secondary data sources that can be used as expertise evidence to automatically extract candidate experts [4]. There are different types of documents

that have been used in the current expert finding systems studies. The most used textual expertise source is publications, which have been used by authors in [36,38,44,48,49,53,58,59,61,62,66–68,70–73,80] as their main expertise source, while publications have been combined with other expertise sources in [22,40,41,43,44,47,55,60,69,75,88,89]. In addition to publications, reference [42] used movies as expertise evidence. Two studies [80,81] used medical documents. Moreover, another two studies [15,77] used employee documents. Also, a bug repository was used as expertise evidence by the authors in reference [36]. Reference [78] used enterprise terminologies in their system as expertise evidence, while reference [79] used transactional data. Additionally, relevance feedback and log repository have been used by authors in reference [74] and reference [35], respectively. The advantage of using documents as expertise sources is the simplicity of extracting up-to-date information about a particular expert. However, their disadvantage is that they cannot contain all the required information. For example, it is difficult to discover junior experts who have no publications. Meta databases are the main store of employees' expertise in organizations. They are used to store personal profiles for employees [4]. For example, a university expertise database saves the profiles of university members with all academic information including personal information, research students, expertise areas, projects, and publications. Among many different expertise sources, faculty and university personal homepages are a valuable and reliable source, since they usually contain relatively well-formatted data [31]. The authors of references [5,26,27,44] used homepages as expertise evidences for their systems. Usually, building and updating such databases is an expensive task, but small organizations are able to keep their databases up-to-date with good quality information [4]. Thus, the shortcomings of this expertise source are (1) the completeness and update of information depends on the employees, and sometimes they are busy or unwilling to carry out this task [71]; (2) the employees need motivation and rewards from their organization to update their profiles regularly; (3) in an organization's databases, the descriptions of expertise tend to be unstructured and general, and it is difficult to seek experts in a specific area of expertise with these descriptions; and (4) it is difficult to update the information in these databases in real time because of the static nature of these databases [1,4]. Additionally, emails have been used as expertise evidence by the authors in reference [87].

2.  Social Networks (SNs)—The rapid development in internet technologies creates a huge amount of user-generated data disseminated on SNs, which is becoming one of the most important expertise sources [53,86]. This user-generated data holds a lot of information such as users' posts, question tags, opinions, activities, experience, etc., which can be valuable for expert finding [71]. SNs are one of the valuable expertise sources since they commonly contain up-to-date information and show meaningful relationships between people. Experts' SNs can be used to extract the effective factors for their ranking and selection (e.g., organisation-related factors, temporal factors, and contextual factors). SNs were used widely as expertise sources by the authors of references [20,21,37,56,57,76,86]. Also, forums have been used by the authors in references [84,85]. Additionally, posts and questions tags have been used as expertise evidences by the authors of references [16,17,19,44,82,83].

3.  Hybrid expertise sources—Recent studies recommended that identifying an expert should not be based on a single expertise source. Thus, current expert finding systems studies integrated heterogeneous expertise sources to enhance the quality of expert finding systems. Several researchers combined different expertise sources, and these are presented in Table 4.

**Table 4.** Expertise sources.

| Channel | Expertise Sources | Studies |
|---|---|---|
| Textual sources | Publications | [36,38,44,48,49,53,58,59,61,62,66–68,70–73,80] |
| | Movies | [42] |
| | Medicine related documents, | [80,81] |
| | Employee documents | [15,77] |
| | Bug repository | [36] |
| | Transactional data | [79] |
| | Enterprise terminologies' | [78] |
| | Relevance feedback | [74] |
| | Log repository | [35] |
| Social networks | Homepages | [5,26,27,44,46] |
| | Emails | [87] |
| | Social network | [20,21,37,56,57,76,86] |
| | Forums | [84,85] |
| | Posts and questions tags | [16,17,19,44,82,83] |
| | Employee documents, publications and projects | [75] |
| Hybrid expertise sources | Proposals, publications | [22] |
| | homepages and social network | [39] |
| | Publications, supervised student theses, course descriptions, and research descriptions | [41,43] |
| | Publications and social networks | [44,47,55,69,88] |
| | Publications and homepages | [40] |
| | publications and citation data | [48] |
| | Projects–homepages | [65] |
| | publications, citations, homepages, Experience | [60] |
| | webpages, project/grant repositories, citation indexes | [44] |

*3.3. RQ3: What Are the Differences between Expertise Retrieval and Expertise Seeking, and What Are the Contextual Factors and Theories That Have Been Integrated with Expert Finding Systems?*

3.3.1. RQ3a: What Are the Differences between Expertise Retrieval and Expertise Seeking?

In this question, the authors differentiate between the expertise retrieval and the expertise seeking tasks using three dimensions—definition, research area, and sociotechnical perspective. As for the definition of expertise retrieval, the authors in reference [68] stated that the expertise retrieval involves the identification of a set of people who possessed the necessary expertise for a specific query. Generally, the expertise of any candidate is based on all documents related to this candidate. This document-based model was seen to be a state-of-the-art approach that estimated the weighted sum of the retrieval scores for all the documents associated with an expert candidate as the measure of the candidate's expertise. After the introduction of the expert finding tasks in the Text Retrieval Conference (TREC) in 2005, there has been an increased interest in the application of the expertise retrieval task in research. While the definition of expertise seeking can be stated as the activity that involves the selection of people as a consultation source based on the information needs. People can be used as sources if they have a higher knowledge than what they usually record in documents. This need for further human-based knowledge can develop in the course of a conversation. These human experts can offer their expertise on various issues [81,82]. Expertise seeking helps in developing experts can offer their expertise on various issues [90,91]. Expertise seeking helps in developing models that determine the manner in which the relevant experts are selected. The relevant models help in identifying the factors that can play a vital role in the expert selection [92].

From research area dimension, some researchers [1,43] considered the expertise retrieval system as a subject of further research in the area of information retrieval. Expertise seeking is considered to be

included in the knowledge management research area, which effectively uses human knowledge in an organisation. Finally, from a sociotechnical perspective, expertise retrieval systems mainly focus on the development of content-based algorithms, similar to a document search. Such algorithms identified all experts by analysing the content of all the related documents. A majority of the research in the expertise retrieval system was dependent on a system-centered perspective, which concentrated on the identification of good matches between the need for expertise and the content of the documents related with the experts. On the other hand, further research was carried out in expertise seeking for determining the techniques used when seeking an expert for a particular task. This addressed the issue of associating humans to an expertise field using a human-centered viewpoint. Furthermore, reference [27] also stated that a majority of the expertise seeking studies focused on determining the manner in which a user search for an expert and seeks the relevant expertise for the specific activity. These studies helped in developing models that can identify the factors that influenced the expert finding. Table 5 summarizes these differences.

**Table 5.** The differences between expertise retrieval and seeking.

| The Concept | Definition | Research Area | Sociotechnical Perspective |
|---|---|---|---|
| Expertise Retrieval | Developing systems that enable expertise seekers to identify and select experts in a particular area. | information retrieval | system-centered perspective |
| Expertise Seeking | Developing models to identify the factors that affect expert selection. | knowledge management | human-centered perspective |

### 3.3.2. What Are the Contextual Factors and Theories That Have Been Integrated with Expert Finding Systems?

Most current expert finding systems retrieved people like documents [4,38], whereas people are unlike documents and are not directly represented as retrievable elements. The representation of experts based on the strength of relationships among a topic and an expert is called representation based on content-based factors [1,25,73]. A challenge in content-based expert finding systems is that the systems need to go beyond document retrieval, as they are required to retrieve entities (experts) instead of documents. According to social capital theory [93], the process of expertise exchange is a social process. For example, the authors in reference [94] found that awareness of "who knows what" is not enough to find the actual access to expertise, and it needs to be supported by strong relationships. Also, the authors of references [95,96] found that employees who have expertise about a particular problem may be unwilling to share their expertise with people with whom they do not share positive affective relations. Thus, it is clear that, besides the degree of expertise extracted from documents, there are other important factors that should be taken into account for expert finding systems development, such as contextual factors [1,25,27,39,71].

The contextual factors are described as factors that influence the manner in which people find and select the experts and can be beyond the various content-based factors [1,27]. The contextual factors included in the information retrieval (IR) studies have highlighted the concept of the context in the field of information science. Hence, context refers to the term that is maximally 'often used', minimally 'often defined', and 'when defined so variously' [97]. The authors in reference [98] stated that context is not an individual factor but a combination of different factors consisting of several elements or aspects. Thus, context can be used as an umbrella term for describing various factors that could simplify things for a specific purpose. The authors in reference [99] described the major concepts of the contexts used in the IR that were investigated by the 2005 SIGIR workshop, i.e., IR in context (IRiX). The researchers attempted to answer the question, which features of context were important to the IR? This workshop led to the determination of different dimensions for the context (i.e., contextual factors), which were (1) task-related factors, like the task level, task type, and goal; (2) seeker-related factors like behaviour,

motivation and knowledge; (3) interaction factors like architecture and technology; (4) system factors like the technology and the interface; (5) object factors (wherein object refers to the experts, movies, or documents, etc.) like accessibility and the quality; and (6) environmental factors that consisting of (a) social factors like norms, community, and roles; (b) physical factors like mobility, temperature, and location; and (c) temporal factors, like the search phase, duration, constraints, and clock-time. Contextual factors are normally identified in expertise seeking research. This study analysed the nine studies in expertise seeking presented in Table 6. This table presents the contextual factors that have been identified in the expertise seeking domain. These factors can be combined with expert finding systems to improve their effectiveness. The major expertise source factors in these studies are the source quality and accessibility. All nine studies investigated the source accessibility, while eight of the studies analysed source quality, either in the form of a single dimension or as a multidimensional factor. The authors of references [92,100] studied the source accessibility and quality as a single dimension; however, some authors [101–107] analysed these two factors as multidimensional factors. Along with the source quality and accessibility, the authors of reference [102] investigated the seeker-source relationship, like the lack of the confront factors. The authors in reference [103] identified various reasons for determining why users preferred to consult their colleagues for some queries rather than seeking the help of the designated internal experts. Their study identified 21 concepts, which were grouped into five separate groups, i.e., four expertise seeking barriers and one context variable for expertise seeking. These four barriers included (1) environment factors of culture and company size; (2) accessibility factors including the unknown experts, expert availability, or the expert willingness to share their knowledge and physical presence; (3) communication factors including the complex answers, expert's ability to transfer their knowledge, expert's adaptation to the user, insufficient expert knowledge, jargon, expert qualifications, their actual experience related to similar issues, and the user's inability to formulate the questions; and (4) personality factors including the loss of face, strong colleague relationships, and a mutual frame of reference. The context factors included the perceived approachability, good starting point, answer quality, shorter answers, and some time constraints. Some of the factors were investigated as antecedents, while others were studied as moderators. The authors in reference [107] analysed time pressure as the moderator factor. The researchers studied the role of various dimensions of the source quality and accessibility and determined how the time pressure affected their significance. Furthermore, reference [100] also studied the manner in which the need for information affected the source accessibility and quality. The authors in reference [102] examined various task-related factors as moderators and also determined how source accessibility and quality were influenced by the task-related factors like urgency, complexity, and task importance. Table 6 explains these factors in detail.

In the past few years, expert finding systems studies have tried to combine contextual factors, derived from expertise seeking to improve their effectiveness. Table 7 summarizes 14 different studies that combined several contextual factors from 68 studies in expertise retrieval systems. The authors of reference [56] combined social factors into an expert finding system developed using a social network context. These social factors included knowledge level, background, experience, and personal preferences of the experts. Also, the authors of reference [88] combined experts' locations and time into their system.

**Table 6.** Contextual factors in expertise seeking.

| Antecedents | Findings | Moderators | Study |
|---|---|---|---|
| Information quality source accessibility | + information quality<br>+ source accessibility | - | [92] |
| Cognitive accessibility, physical accessibility, relational accessibility, reliability, relevance | + quality<br>+ accessibility | | [96] |
| social accessibility of expertise providers, technological accessibility, and awareness | + social accessibility<br>+ technological accessibility<br>+ awareness | - | [92] |
| Quality and accessibility (Physical accessibility, cognitive accessibility, relational accessibility, relevance technical quality(reliability)) | Without time pressure:<br>+ Physical accessibility<br>+ Cognitive accessibility<br>+ Relational accessibility<br>+ Relevance<br>+ Technical quality(reliability))<br>Under time pressure:<br>+ Physical accessibility<br>+ Cognitive accessibility<br>+ Relational accessibility<br>+ Relevance<br>− Technical quality(reliability) | time pressure | [98] |
| Relationships and awareness | + Positive relationships<br>(No effect) Negative Relationship<br>+Awareness | - | [97] |
| Quality, accessibility | Both quality and accessibility are important. Information need can function as a contingency factor such that the relative importance of quality over accessibility shifts with information need. | Information need | [91] |
| Willingness to help, network ties self-identified expertise, awareness of other resources, communication skills | 'Willingness to help' and 'network ties to the respondent' were more important for knowledge allocation. 'self-identified expertise' and 'awareness of other resources' were more important for knowledge retrieval. There was no significant difference in the importance of 'communication skills' attribute. | | [95] |
| source factors: Source quality, access difficulty, communication difficulty, seeker-source relationship: lack of confront | + source quality<br>+ access difficulty<br>+ communication difficulty<br>+ importance<br>− urgency<br>− complexity | task factors: importance, urgency, complexity | [93] |
| Environment factors Accessibility factors Communication factors Personality factors Context factors | | | [94] |

(+ means positive effect and − negative effect).

**Table 7.** Contextual factors in expertise retrieval systems.

| Dimension | Factors | Research collaboration | Reviewing (Academia) | finding similar experts (Academia) | R&D task | Not Specified | Enterprise | Social Networks | Online Knowledge Communities | Reference |
|---|---|---|---|---|---|---|---|---|---|---|
| **Expert-related factors** | Topic of knowledge, familiarity, reliability, availability, perspective up-to-dateness, approachability, cognitive effort, contacts, physical proximity, saves time | | | √ | | | | | | [27] |
| | connectivity | √ | √ | | | | √ | | √ | [20–22,39] |
| | quality | √ | √ | | √ | | √ | | √ | [20,21,25,35,39] |
| | Relevance | | | | √ | | √ | | √ | [17,22,25] |
| | user reputation | | | | | | | | √ | [17] |
| | authority | | | | | | √ | | | [17] |
| | accessibility | | | | | | √ | | | [35] |
| | productivity | | | | | √ | | | | [22] |
| | time to contact an expert and the knowledge value | | | | | | √ | | | [43] |
| | Trust | | | | | √ | | | | [25] |
| **Community factors** | Number of documents Number of authors for each community | | | | | | √ | | | [54] |
| **Temporal factors** | Location and Time | | | | | | | √ | | [88] |
| **Organizational factors** | Organizational structure and Media experience | | | √ | | | | | | [27] |
| **Social factors** | Experience, background knowledge level and personal preferences of experts | | | | | | | √ | | [56,57] |
| **User motivation factors:** | Most qualified Friend of friend Birds of a feather Social exchange follow the crowd | √ | | | | | | | | [55] |

In academia, various factors were combined for developing expert finding systems. The authors in reference [68] combined the community factors, namely the number of documents and the authors for every community in their system. Furthermore, for a research collaborative activity, the authors in reference [55] incorporated user motivation factors with their expert finding system. They used the following motivation factors: qualification, a friend of a friend, social exchange, birds of the feather, and following the crowd. The authors of references [20,21,39] combined the connectivity, relevance, and the quality to their respective expertise retrieval systems. For determining similar experts' tasks, the authors of reference [27] integrated expert-related factors, namely familiarity, availability, the topic of knowledge, perspective, reliability, up-to-datedness, approachability, contacts, cognitive efforts, physical proximity, and time-saving, along with the organisation-related factors media experience and organizational structure. Furthermore, for research and developmental tasks, the authors in reference [25] combined various expert-related factors, namely trust, quality, and relevance. In an enterprise context, the authors in reference [35] combined accessibility and quality factors into the expert finding system, while the authors in reference [22] integrated connectivity, productivity, and relevance factors into their proposed system. In the KSCs, the authors in reference [17] integrated authority, user reputation, and relevance into their system.

The popular expert-related contextual factors, as shown in Table 7, indicate that the most frequently used factor was quality, followed by connectivity and relevance.

To date, only one paper out of the 68 expert finding system–related studies developed an expert finding system based on the theories. In their study, the authors of reference [55] applied the Multi-Theoretical Multi-Level (MTML) framework, which used different results acquired from the various social theories and the network formation systems used in communities for constructing a catalogue of several individual motivations, namely discovering new people (novelty), exploring the existing resources, collaborating with people having similar characteristics (homophily), and showing a faster response. The researchers used various theories in their framework, including balance, self-interest, social exchange, homophily, and contagion theories.

## 3.4. RQ4: What Are the Methods That Have Been Used for Expert Finding Systems?

In an expert finding system, the relationship between query topics and people is commonly known as the modelling task. Expert finding models have three main components— (1) candidate, which represents a person who might be an expert, (2) document, which refers to the heterogeneous expertise resources, and (3) topic, which is a specified domain [4]. In the literature, there are several models that have been developed to capture the relations between query terms and expert candidates. These models are discussed in the following subsections, and the previous studies that used these models are represented in Table 8.

### 3.4.1. Generative Probabilistic Models

These models estimate the relationships between query topics and people as the probability of generating a particular topic by a given candidate. This type of model is the first edition of the Text Retrieval Conference (TREC) Enterprise track, and it considered a popular class for expertise retrieval. This is due to TREC's good empirical performance and prospective for combining different extensions in a transparent and theoretically sound manner. The ranking process of the candidate experts are based on the probability of person e being an expert on query q (*P(e|q)*). Commonly, there are two methods for estimating this likelihood, and these define the two core types of generative models. The first type is the candidate generation model, which can be generated directly according to the query topic q and the probability model finding the candidate expert e. The second type of generative models is a topic generation model, based on Bayes' Theorem, and it can be represented as follows:

$$p(e \div q) = (rank)p(q \div e) \times p(e) \tag{1}$$

where P(e) is the likelihood of a candidate and $P$(q) is the likelihood of a query [1]. Since $P(q)$ is a constant (for a given query), it can be disregarded for the purpose of ranking. Thus, the likelihood of a candidate expert e being an expert given query q is proportional to the probability of the query given the candidate ($P(q|e)$), weighted by the a priori belief that candidate e is an expert ($P(e)$) [108].

Generative probabilistic models contain different methods such as a document-based model. Document-based methods (also called query-dependent methods) first find the related documents to the query and then rank candidates associated with these documents based on an integration of the relevance score of documents and the degree of a person's association with that documents. Therefore, individuals are represented based on a weighted set of documents. Generative probabilistic models have been used by several researchers, as shown in Table 8. Reference [40] used content-based methods and explored different techniques for computing similarity between two researcher profiles, namely

(1) Okapi BM25 (OKAPI), where a vector of terms associated with researcher's content was used to represent the researcher profile. In order to obtain the similarity between two profiles, this technique considered one profile as the query and the second profile as a document.
(2) KL Divergence (KLD): In this technique, a probability distribution is used to represent a researcher profile. For example, a multinomial distribution can be used to model the term counts in a researcher's associated documents and to count the similarity between two probability distributions. KullbackLeibler divergence is used.
(3) Probabilistic Modelling (PM): Latent Dirichlet Allocation is a widely used tool for topic modelling in unsupervised clustering of data and exploratory analysis. For example, when researchers have a desire to work on multiple related areas, sometimes it is more suitable to model their profiles as topic mixtures rather than a single multinomial distribution.
(4) Trace-Based Similarity (REL): used density matrices to represent vector subspaces for modelling concepts. Additionally, reference [27] used content-based similarity to develop a model for finding similar experts in a university.

Also, reference [109] used generative probabilistic and similarity models to propose Beneficial Collaborator Recommendation (BCR). Also, reference [77] used generative language models for group finding task from heterogeneous document repository.

### 3.4.2. Discriminative Probabilistic Models

Discriminative models are an essential class of probabilistic models with a strong statistical base. There are theoretical results that indicate that, according to the increase in training set size, these models have a lower asymptotic error [110]. When some training data is available, discriminative models have been preferred over generative models in several information retrieval applications such as text classification and information extraction [111]. The success of generative models mainly relies on the validity of the model assumptions. These assumptions are occasionally too strong, such as the independence assumption of term distributions. Discriminative models usually make fewer model assumptions and prefer to let the data speak for itself.

Initial work on applying discriminative models in information retrieval dates back to the early 1980s, when the maximum entropy method was examined to get around the term independence assumption in generative models [112]. Recently, discriminative models have been applied to retrieval problems such as homepage finding [113], e-mail retrieval [114], and question answering [115]. One of the most significant interests in the IR community in recent years is learning to rank for ad-hoc retrieval [116]. Maximum learning to rank models are discriminative in nature, and they have been presented to be superior to generative language models in ad-hoc retrieval. Benchmark data sets such as LETOR [116] are also available for the study of learning to rank. With respect to expert finding, when generative models determine the relations between people and documents, the relations tend to be ambiguous in the TREC Enterprise settings as well as in numerous realistic scenarios. In generative models, the number of relations signals is very limited, but the method of combining them is regularly

heuristic, and shortages are clearly justified. An additional significant component in generative models is document evidence, which potentially contains many document relevance features. These features contain document authority information such as the PageRank, indegree, and URL length [117].

### 3.4.3. Voting Models

These models are based on methods from the field of data fusion [118]. Data fusion methods are concepts of integrating evidence from heterogeneous sources, rankers, or representations and have a long history in information retrieval [119]. The use of data fusion methods in expert finding is built on the intuition that the documents associated with candidate experts ranked with regard to a query can be seen as 'voting' for candidate experts. The voting methods for expert search aggregate scores from the same system across expert candidates instead of combining different systems' scores on a single document as in metasearch [1]. The authors of reference [118] proposed the simplest method in voting models. Their method assumes that documents provide binary votes given that they appear among the ranked documents returned by a document search engine. Candidate experts are ranked by the number of retrieved documents associated with them. Reciprocal Rank (RR) and CombSUM are examples of data fusion methods used for expert flinging systems. The authors of [69] used CombSUM and CombMNZ to combine different sources of expertise derived from the textual contents, the citation patterns for the community of experts, and an expert's profile. Also, the authors of reference [60] and reference [26] developed their models based on the weight that extracts the candidate's expertise from their documents. The authors of reference [60] calculated the total weight as the summation of publication, citation, and editor weight, while the authors of reference [26] focused on the distance weight. It is about the proximity between query terms and candidate names in the related documents, and it will be higher whenever the candidate name appears closer to the query term.

### 3.4.4. Network-Based Models

Currently, referral webs and social networks are used as basic channels for finding experts' information [4]. Network-based models use graphs to represent the main elements of expertise retrieval models. These graphs can be built in two ways. First, graphs represent documents and candidate experts as nodes, and their relationships are viewed as edges. Second, nodes are used to represent candidate experts only, and edges are used to represent the relationships between the candidate experts [1]. There are many algorithms that are commonly used in network-based models, such as HITS and PageRank algorithms and random walk propagation algorithms [4]. HITS [120] and PageRank [121] were both suggested in 1998 and used in web searching to discover the importance of a web page. The algorithms' concepts and extended approaches are similar to the Internet in structure when using expert finding in expert networks. Candidate experts or documents can be seen as web pages on the Internet, where hyperlinks among web pages represent the candidate–candidate relationships and the candidate–document relationships of the expert finding model [1]. An additional effective ranking algorithm is the PageRank algorithm, which has been effectively used by Libra, Microsoft's scholar search engine [4]. The authors of reference [47] proposed a SWAT framework for team recommendation based on network-based models. SWAT is based on a model categorized into three parts—individuals, expertise areas, and social dimensions. The three parts are captured using the competence graph, the social graph, and the history graph, respectively. Also, the authors of reference [67] proposed a graph-based regularization method using Enhanced Bl and Co-Ranking. Enhanced BL is an enhanced model for expertise ranking. It integrates the co-authorship graph with query dependent community on top of a BL model. Moreover, the authors of reference [84] adapted the PageRank algorithm for online help seeking as an expertise rank algorithm in online forums. Their algorithm considered the reputation of users in Java discussion forums. PageRank has been adapted to expertise rank extended to ExpRank-COM to measure reputation. Additionally, the authors in reference [18] modified the PageRank algorithm to ExpertRank to assess an expert's authority so that it decreases the effect of certain biasing communication behaviour in online communities.

Propagation models are also network-based, the authors of reference [24] used a graph-based method and developed the grouping-based Steiner algorithm (GrpSteiner) to find experts in social networks. Most of the generative probabilistic models and voting models cannot mine the hidden information that can be found in social networks and that cannot be mined from the documents, and they are 'one-step' relevance models [4]. The authors of reference [46] deployed entity linking, relatedness, and entity embeddings to create a novel WEM profile for academia experts. They built their model based on a weighted and labeled graph drawn from Wikipedia.

### 3.4.5. Hybrid Methods

authors of references [56,57] combined content-based recommendation algorithms into a social network–based collaborative filtering system to find experts from social networks. Also, the authors in reference [55] proposed a model based on computational techniques from social network analysis and representational techniques from the semantic web to facilitate combining and operating data from heterogeneous sources. The authors in reference [68] combined a document-based model (generative probabilistic models) and community-sensitive AuthorRank in their system. Additionally, the authors in reference [42] combined cluster-based expert selection with a collaborative filtering algorithm to solve the problems of scalability and noise caused by the collaborative filtering algorithm. The authors in reference [17] combined a cluster-based language model (CBDM), Bayesian smoothing with the Dirichlet prior technique, and a vector space model to develop a category-based and question-dependent approach for finding experts in question and answer communities. Moreover, the authors in reference [70] integrated the Dempster-Shafer theory of evidence and Shannon's entropy with a multisensor data fusion framework to develop an expert finding system in academia. Also, the authors in reference [16] proposed a learning framework to predict the ranking of experts in question and answer communities. In their framework, they combined probabilistic dynamic expert profiling, topic modelling, and a document-centric model. In online communities, the authors in reference [82] presented a hybrid method for expert finding. Their method was based on combining content analysis (based on concept map) and social network analysis (based on PageRank algorithm) and social network analysis combined with email data mining to develop a personalized expert recommender tool [87]. Moreover, reference [79] proposed a model based on integrating Term Frequency and Inverse Document Frequency (TF-IDF) and the document language modelling approach to find experts in enterprise. Additionally, reference [25] proposed a context-aware researcher recommendation system based on combining content based, the social network-based and the sum method. In order to find experts for a certain topic, the authors of reference [88] combined topic-aware model (TAM) and context-aware model (CAM) to propose a topic-specific contextual expert finding method. Also, reference [53] combined social network analysis and semantic concept analysis to propose a network based researcher recommendation approach to enhance the efficiency of personalized researcher recommendation. The authors of reference [54] proposed a novel model called Friend Recommendation Model (FRM) by integrating Filtering Out Model (FOM) and Linear Combination Model (LCM).

### 3.4.6. Others

Expert search is closely related to document search and other IR retrieval tasks; both traditional and modern IR methods have been applied to expert search, such as fuzzy logic, the vector space model, cluster-based retrieval, and big data. The authors in reference [74] used fuzzy logic in expert finding systems. In their model, they created fuzzy relevance profiles using TF-IDF analysis. They combined three profiles into a unified index reflecting the semantic weight of the query terms related to the task, user and document to select the most relevant documents and experts related to the query topic. The unified index was used to select the most relevant documents and experts related to the query. The profiles integrated by fuzzy rule–based summarization. Furthermore, reference [66] proposed model based on Support Vector Machine and they used Random Forests for author names disambiguated. The authors in reference [61] developed two algorithms in Java to calculate the best

distance for their model. They studied the problem of finding a team of experts with required skills and who have the minimal communication cost from a social network. Moreover, reference [15] adopted the fuzzy linguistic method to build the expert profile and used the fuzzy text classifier to get the relevant degree of the document to each area of knowledge. Reference [45] developed an expert finding system based on the Dempster-Shafer theory. This theory was used to combine different expertise sources. Also, the authors of references [5,51] used web mining methods to propose multifaceted web mining heuristic tool to find people of desired expertise and to recommend scholars, respectively. Also, the authors of references [62,63] used facility location analysis to propose an optimization framework to find an optimal group of experts for a given multi-aspect task/project. In addition to that, the authors in reference [85] proposed a novel algorithm based on the techniques provided by the WordNet dictionary to find experts in forums communities. The techniques used are text mining techniques and semantic similarity function. Additionally, reference [81] proposed a context adaptive algorithm to find the optimal expert for patients with a specific context. Furthermore, K-Nearest-Neighbor was used by the authors in reference [36] to propose a new approach called DREX to develop expert finding. Also, ontology has been used for expert finding systems. This is a formal explicit description of classes in a particular area with set of properties to describe the various features and attributes. For example, the authors of references [50,78] developed ontological models for expert finding systems. Moreover, recommendation systems techniques have been used for expert finding. For example, the authors in reference [59] proposed a content-based recommender system to select reviewers to evaluate research proposals or articles. They developed a comprehensive algorithmic framework based on different IR techniques. Also, reference [83] proposed a recommendation algorithm called collaborative expert recommendation (CER) for expert finding based on collaborative filtering. On the other hand, big data techniques and tools have also been used for expert finding systems. For instance, reference [39] used big data to adopt an intelligent recommendation method to select appropriate experts for peer review in online research community. The authors of reference [22] proposed a social network-empowered research analytics framework (RAF) to select research projects and reviewers. The authors of reference [20] developed a novel model based on big data analytics with MapReduce to recommend experts in scientific communities.

**Table 8.** Expert finding systems methods.

| Methods | References |
| --- | --- |
| Generative probabilistic models | [27,40,77,109] |
| Discriminative probabilistic models | [31] |
| Voting models | [26,60,69] |
| Network-based models | [18,24,46,47,67,84] |
| Hybrid methods | [16,17,25,42,53–57,68,70,79,82,87,88] |
| Others | [5,15,20,22,36,39,45,50,51,59,61–63,74,78,81,83,85] |

*3.5. RQ5: What Are the Datasets That Have Been Used for Expert Finding Systems?*

Expert finding systems are IR systems where interactions start with a user sending a query to the system, and then the system retrieves a list of experts that hopefully are relevant to the user's query. Normally, an IR research evaluates the system efficiency using a test data collection (datasets). The most popular evaluation criteria that are generally used are mean reciprocal rank (MRR), mean average precision (MAR), and precision after ranking N people (P@N). Earlier component of a test collection contains (1) a collection of documents, (2) a set of topics, as queries in the collection, and (3) a set of relevance judgments, also called a query relevance set. To test the implemented expert finding systems, developers can load the documents from the dataset (text collection) and send the topics to their systems one by one. Then the documents retrieved by the developed systems can be examined by their occurrence in the text collection. The efficiency of the developed models can be displayed by comparing them with baseline retrieval systems [4]. Expert finding systems uses the

existing test collections of IR [4,122] and additional test collections shown in Table 9. The table shows the test collections, their characteristics, and the previous studies that used them. A particular test collection can be used for expert finding or expert profiling or both of them.

The internal documentation of World Wide Web Consortium (W3C) is generated from public W3C sites and produced the W3C collection which was used in TREC 2005. Table 9 displays the statistics of W3C in TREC 2005. Documents in the dataset hold intranet pages, papers, email, etc. The predefined candidate experts are the participants in working groups, and the search topics are these working groups in W3C. This technique has some obvious limitations. For instance, the document dataset may hold data on other experts that are not on the set. Then, in TREC 2006, 55 topics are chosen for the final set [4]. In TREC 2007, the test collection is developed into a snapshot of the Australian Commonwealth Scientific and Industrial Research Organization (CSIRO). A main difference between the CSRIO and the W3C is that CERC does not offer a predefined set of expert candidates, and the number of experts on each topic is fewer. It is worth stating that the expertise judgment is made by the authors themselves with other encoded knowledge [123].

The academic test collections that provided for expert finding have been developed and made available for the public. For example, the UvT [124] and its enhanced version TU [41] contains employees of Tilburg University as candidate experts. The university official webpages used by UvT have some interesting characteristics and can be considered valuable resources of expert relevant documents. Commonly, faculties and scientists in universities or colleges are seen to have certain specialties on specified knowledge fields. Additionally, personal profiles on official webpages are mostly maintained by the individuals themselves. These self-introductions are updated when they make any progress on their research and knowledge learning. The TU dataset is currently the largest dataset accessible for benchmarking academia expert finding systems. It contains both Dutch and English documents, and it has five different ground-truths—GT1 to GT5. GT5 is considered the most recent and complete ground-truth [46]. Additionally, CiteSeer and DBLP offer the information of computer scientific literatures on main journals and conference proceedings. DBLP is the most commonly used data set in expert finding systems research (see Table 9). CiteSeer has developed a new version, CiteSeerX, which aims to provide more resources such as algorithms and metadata [33,125]. CiteSeer test collection provide huge sources of experts and their associated documents for additional research and test data collection. It has been used by references [47,66]. Additionally reference, [88] combined CiteSeer and Twitter test collections. They extracted the title, abstract and citation relationship from CiteSeer and user ID, posts, reading and commenting information from Twitter. The authors of reference [47] used different data sets, namely: DBLP, Microsoft "Academic Search", ACM Digital Library, Google Scholar, Springer's Digital Library, Google Maps API, CiteSeerX, IEEE Xplore, Academia.edu, and Facebook to combine data from different data sources. Reference [47] used DBLP to retrieve individuals' information, publications and venues. These data are incomplete because the information about expertise areas is ambiguous and the data does not hold much information about the individuals and the relationships between them (or publications). Therefore, they expanded the information about individuals and developed a set of wrappers to retrieve the required data from Academia.edu and Facebook (they are public (and academic) social networks). Facebook requires data owner authorization to extract the publicly available and private data. The study extracted information from academic indexing services, namely Microsoft Academic Search and Google Scholar (web search engines for academic publications), to identify the research interests and the individual's affiliations. In order to identify the geographic locations of the retrieved organizations, they used the Google Maps API. Also, the study extracted additional information such as citation counts, abstracts, and topics from different research databases, namely ACM Digital Library, Springer's Digital Library, CiteSeerX, and IEEE Xplore.

In addition to the previous test collections, the authors in reference [6] presented a novel test collection called Lattes Expertise Retrieval (LExR). It was developed to offer a unified benchmark for researchers for expert finding and profiling in academia.

Data has also been extracted from the Wikipedia data set. It is a document repository that holds articles, and each article only defines a single topic. In each article, the title is a concise phrase called a concept. In Wikipedia, categories are organized in a hierarchical structure, and this make it a good ontology to be used for extracting semantic correlations between diverse concepts [56]. Additionally, question and answer websites such as Stack Overflow, Yahoo! Answer, and Ask Me Help Desk are used to extract data sets for expert finding systems. The authors of references [16,19,83] used Stack Overflow (which currently contains 16 million questions 2019), while reference [17] used Yahoo! Answer. Ask Me Help Desk has been used by the authors of reference [85]. Ask Me Help Desk contains more than 500,000 questions and more than 2.5 million answers. It was ranked as one of the best Top 13 Sites for answering questions. Social networks have been widely used in expert finding systems as data sets to extract information for finding experts. In addition to Twitter, Facebook, and Academia.edu, Scholarmate has been used by the authors of references [20,22,25].

**Table 9.** Data set characteristics.

| The Collection The Collection | Environment | Year | Inter-Organization | Inter-Area | Expert Profiling | Expert Finding | No of Documents | Studies Used the Test Collection |
|---|---|---|---|---|---|---|---|---|
| TREC | enterprise | 2007 | | ✔ | | ✔ | 370,715 | [27,74] |
| | | 2006 | | ✔ | | ✔ | 331,037 | [77] |
| UvT | academia | 2006 | | ✔ | ✔ | ✔ | 36,699 | [40,43] |
| TU | academia | 2008 | | ✔ | ✔ | ✔ | 31,209 | [46] |
| DBLP | academia | 2008 | | ✔ | | ✔ | ≥2,300,000 | [5,24,45,49,54,61,67–70] |
| CiteSeer | academia | 2008 | | ✔ | | ✔ | ≥750,000 | [47,66,88] |
| LExR | academia | 2015 | ✔ | ✔ | ✔ | ✔ | 11,942,014 | [6] |
| Twitter | Social network | 2009 | | ✔ | ✔ | ✔ | - | [88] |
| Microsoft Academic Search | academia | 2016 | ✔ | ✔ | ✔ | ✔ | >220,000,000 | [47] |
| ACM Digital Library | academia | 1950 | ✔ | ✔ | ✔ | ✔ | >1,000,000 | [27,38,47,62,63] |
| Google Scholar | academia | 2004 | ✔ | ✔ | ✔ | ✔ | >389,000,000 | [47,50] |
| IEEE Xplore | academia | 1988 | | ✔ | ✔ | ✔ | >4,500,000,000 | [47] |
| Facebook | Social network | 2006 | ✔ | ✔ | ✔ | ✔ | - | [47] |
| Scholarmate | Social network | 2007 | | ✔ | ✔ | ✔ | - | [20,22,25] |
| Academia.edu | academia | 2008 | | ✔ | ✔ | ✔ | >21,000,000 | [47] |
| Wikipedia | academia | 2001 | | ✔ | ✔ | ✔ | >40,000,000 | [56,57] |
| Stack Overflow | question and answer website | 2008 | | ✔ | ✔ | ✔ | - | [16,19,83] |
| Yahoo! Answer | question and answer website | 2005 | | ✔ | ✔ | ✔ | - | [17] |
| Ask Me Help Desk | question and answer website | 2003 | | ✔ | ✔ | ✔ | - | [85] |

In addition to the well-known data sets shown in Table 9, there are other data sets that have been used for expert finding systems. For example, reference [53] used all papers published for the Pacific Asia Conference on Information Systems from 2007 to 2009, while the authors of reference [60] used the publications from the Journal of Universal Computer Science to identify experts. Moreover, reference [81] used a real-world breast cancer diagnosis data set, and reference [15] used real data for experts in enterprise. Also, reference [79] used enterprise resource planning (ERP) systems as a source of expertise evidence. UNESCO research projects data has been used by the authors of reference [78] to discover experts. The authors of references [26,31,39,42,55] used extracted information

from university websites. Additionally, data sets from online forums have been extracted; for example, the authors of references [18,82] extracted data sets from the Microsoft Office Discussion forum and Oracle corporation forum, respectively. Also, the Mozilla Firefox open bug repository has been used by the authors of reference [36].

## 4. Research Findings on Identified Gaps

This section presents research gaps and future work derived from the results of research questions and discussion presented in Section 4. This paper identifies the following gabs for future researchers in expert finding systems.

1.  Academia domains versus others—Table 3 shows that, from the six main domain categories in this study, the majority of the selected papers (65%) used expert finding systems in academia, which is developed for different tasks like finding research collaborators, paper reviewers, supervisors, similar experts, group of experts, and university–industry collaborators. Although using expert finding systems in academia is promising, integrating these tasks into one comprehensive system for the whole university is important. This system can assist universities in managing their knowledge assets, which is important for innovation. It also assists universities in finding the gap in specific areas, can support expert seekers by providing them to find an expert based on their knowledge, and can encourage them to seek knowledge and help. Additionally, it can assist the experts themselves by appreciating their efforts when they help people and share their knowledge; this leads to an increase in their reputation and widens their networks. Expert finding systems research have not attracted much attention in other domains. For example, in medicine, it can be used to find pharmacists, medical laboratory staff, dentists, and other medical staff. Developing these systems will help people all over the world to find an expert for their specific needs at any time.

2.  Expertise sources availability—According to the critical analysis of the selected studies, the authors found that the availability of expertise sources is a main challenge in the current studies of expert finding systems due to the following issues:

    a)  Information privacy—System developers and organization owners must clearly identify the benefits of their systems. If users perceive value from expert finding systems, they will use these systems and supply the required personal information. Additionally, privacy-preserving techniques should be developed, and users should be given fair control over the storage and usage of their information [75].

    b)  Data integration—In general, integration of different expertise sources is a challenging task, and there are four problems in data integration need more effort from researchers in the future, which are the following: (1) nicknaming, referring to the concept of different names in multiple networks for the same person; (2) name ambiguity, referring to individuals in social networks that have very similar or exactly the same name and a name may refer to different people; (3) multiple references, or finding more than one profile for the same person in a network; and (4) local access to profiles, as sometimes it is difficult to access the whole profiles in a network simultaneously. Thus, there is a need for crawling methods to access the profiles.

    c)  Information updateness and completeness—integrating several data sources is very important in order to guarantee the completeness of expertise-related evidence. For example, in the academia domain, various expertise sources could be used, such as homepages, publications, research descriptions, course descriptions, social networks, project/grant repositories, supervised student theses, citation indexes, and movies. This information can be collected from staff homepages if the homepages are updated. A lot of this information has a shorter lifetime, and incomplete data can lead to wrong decisions.

Thus, there is a lack of research for developing motivation frameworks for organizations to motivate their employees to update their profiles regularly.

3.  Expert finding systems methods—there is a need for developing sophisticated algorithms to integrate several expertise related evidences. For example, in a university, the expert profile is an important source of information due to the high quality of the data, but the data is not complete and is not usually updated. Furthermore, a social network also consists of a huge volume of data, but generally, the data quality is poorer than that of university data repositories. Hence, universities must integrate these information sources.

4.  Data sets—The current data sets for expert finding systems have been developed for a particular environment—for example, UvT, TU, DBLP, CiteSeer, and LExR—were developed for academia. TREC was developed for enterprise, and Yahoo! Answer, Facebook, and Twitter datasets were developed for social networks. There is a need for research to develop data sets that combine information from academia and social networks and from enterprise and social networks because they can complement each other, and the completed benchmark data sets can encourage expert finding systems developers to develop different systems. The benchmark data sets reduce the effort needed for solving information updates problems, information completeness, data cleaning, and name ambiguity problems. Additionally, many of the data on the web (not developed as a well-known data set) are either not updated or suffer from fake entries, and sometimes developers face a lack of access to resources. Therefore, developing benchmark data sets will reduce these challenges.

5.  Combining expertise retrieval and expertise seeking—Expertise retrieval systems and expertise seeking are concerned with finding experts, but expertise retrieval involves developing algorithms for identifying a set of people who possess the necessary expertise for a specific query based on all documents related to this candidate. Expertise seeking can be described as the the selection of people as a consultation source based on information needs. It is needed for further human-based knowledge. Expertise seeking helps in developing models to identify the factors that could play a vital role in an expert's selection based on a human-centred viewpoint. These factors have been called contextual factors in previous research. A majority of research in expertise retrieval systems is dependent on a system-centred perspective, which concentrates on the identification of good matches between the need for expertise and content of documents related to experts. Integrating contextual factors with expertise retrieval systems can improve their effectiveness. Recent research has tried to combine these factors with expertise retrieval systems. Based on this study, the most factors that have been combined from expertise seeking to expertise retrieval are expert-related factors such as quality and accessibility. From the 14 studies in Table 7 that combined contextual factors with expertise retrieval, 10 studies combined expert related factors and the reset combined social, organizational, motivational, and community factors. Accordingly, expert-related factors are the most important factors in expertise retrieval. A majority of the current expert finding systems focused on the global notion of expertise. Due to the difficulties noted in the determination of the accepted definition of an expert along with the heterogeneity of the expertise sources, the complete data about the expert cannot be acquired. Hence, one needs to consider a personalised view of the expertise based on user-specific information requirements. Therefore, there is a need for researchers to develop suitable models based on a theoretical foundation that can examine the manner in which contextual factors influence expert selections in every domain. In the academic domain, researchers must develop a contextual-based model for each particular task, since it can be difficult to develop a common contextual-based model for all of a university's tasks. These models can be integrated with content-based models to improve the performance of expert retrieval [16,25,27]. Previous studies have shown that there is a lack of information system research based on theoretical background for expert finding systems, such as developing user satisfaction and system success models. Additionally, there is a need for qualitative research in expertise seeking to explain how human-related factors influence the

expert finding problem. There is a further need to develop novel expertise seeking models for particular tasks and domains. Finally, several factors affected the source selection in expertise seeking; the most frequently used factors are quality and accessibility, but their interactions are not yet understood. Many studies have investigated task- and seeker-related factors and determined the manner in which source accessibility and quality were affected by the various moderating factors. These factors moderate the effect of source quality and accessibility.

## 5. Threats to the Validity

There are three threats to validity (TTVs) that are usually associated with systematic literature reviews or survey articles—(1) coverage of relevant studies, (2) biasness on the selection of relevant studies (internal validity), and (3) data extraction inaccuracy. This research acknowledges these shortcomings by strategizing to reduce the risks of these TTVs. The three threats of validity were resolved in three separate strategies.

1.  Construct validity was confirmed by an automated and manual (snowballing) search from the beginning of data collection to diminish calculated risks. In order to further control this TTV, the main steps of analysis and additional QAs were carried out complementing clear selection criteria.
2.  Internal validity was addressed by a combination method of automated and snowballing search, which was used for more comprehensive selection approach in order to reduce biases in paper selection. Every study went through strict selection protocols after its extraction from the identified databases.
3.  External validity was diminished with a generalizability of results by including a 10-year timeframe in expert finding systems and expertise seeking studies. The moderate number of studies (96 studies) could be an indicator that this SLR is capable of maintaining a generalized report.

## 6. Conclusions

In this study, we have provided an overview of various expert finding systems concepts. In order to understand these systems, we determined our specific scope by formulating five different research questions. These research questions were mainly formulated to find out the existing domains of expert finding systems, expertise sources, methods, datasets, the differences between expertise retrieval and expertise seeking, and the factors and theories that have been combined to create these systems. This review studied research papers published between 2010 and 2019. After employing multiple processes, we selected 96 studies that focused on expert finding systems and expertise seeking. The other studies were eliminated from this review as they could not satisfy the various inclusion criteria and did not display the desired quality. This study identified research methodologies that were used in the studies. The majority of these studies involved a quantitative method (94%), followed by review (4%), and qualitative and mixed methods (1%, each). Also, this study presented different domains that have applied expert finding systems for different purposes. After data analysis, all selected studies were developed in six domains, with a majority in the academia domain (64%). Other domains, like enterprise, online KSCs, social networks, online forums, and the medical domain, had less attention. Additionally, this study described the expertise sources that have been used for expert finding systems, which were divided into three main categories—textual sources, social networks, and hybrid expertise sources. Moreover, it presented the methods that have been used for models building and placed these into six categories—generative probabilistic models, discriminative probabilistic models, voting models, network-based models, hybrid methods, and other models. Also, the datasets that have been used for evaluating expert finding systems were discussed; they were mainly used in three environments—enterprises, academia, and social networks. Moreover, the differences between expertise retrieval and expertise seeking were discussed, and the contextual factors and theories that have been integrated into expertise retrieval were identified. Finally, gaps in the current studies were identified.

The contributions of this study are (1) it identifies different domains of expert finding systems, (2) it differentiates between expertise seeking and retrieval, (3) it identifies the contextual factors that have been combined to expert finding systems to improve their effectiveness, (4) it describes the state-of-the-art methods and datasets in expert finding systems, and (5) it identifies five main research gaps for future researchers in the expert finding systems domain. In conclusion, after reviewing all the existing studies, we state that this effort would be very helpful for academics, practitioners, and developers of expert finding systems. Expertise retrieval systems have garnered a lot of attention from researchers, and this review could help them identify the gaps and loopholes in this area of research and achieve an overview of existing studies to design their own research.

**Author Contributions:** Conceptualization, O.H.; methodology, O.H. and R.A.A.; validation, O.H., N.S., and R.A.A.; formal analysis, O.H.; investigation, O.H. and N.S.; resources, O.H., N.S., R.A.A., S.A., and A.H.; data curation, O.H.; writing—original draft preparation, O.H.; writing—review and editing, O.H., N.S., R.A.A., S.A., and A.H.; supervision, N.S. and R.A.A.

**Funding:** This paper funded by Prof. Dr. Naomie Salim, Omayma Husain and Prof. Dr. Rose Alinda Alias Universiti Teknologi Malaysia High Impact Research Grant (Vote # Q.J130000.2455.04G78)).

**Acknowledgments:** We would like to thank Syed Norris Hikmi for his valuable inputs and contribution to this paper. We also acknowledge Ministry of Higher Education, Sudan, and University of Khartoum, Faculty of Mathematical Science, Sudan, for sponsoring Omayma Husain in her PhD Program at Universiti Teknologi Malaysia.

**Conflicts of Interest:** The authors declare no conflict of interest.

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
