# Peer review of "Expert Finding Systems: A Systematic Review"

_applsci, doi:10.3390/app9204250_

Round 1

Reviewer 1 Report

The paper addresses an important challenge of expert finding.

An extensive set of literature has been reviewed.

However, weak aspects of the paper include the very general level of discussion and unclear criteria of selection of number of factors. These include the domains, expertise seeking factors, contextual factors, etc.

At this very general level of discussion it is unclear who is the targeted reader and what is the value of the study for her/him.

Furthermore, some methods e.g. ones in Tab. 4 could be described in more detail.

Author Response

Hello Dear reviewer 1,

First of all, I want to thank you for your valuable comments. All the comments have been revised.  Kindly find the required modifications in the following table and the amended version of the paper attached below. Please do not hesitate to contact me if there is any misunderstanding 

Comments

Modifications

Weak aspects of the paper include the very general level of discussion and unclear criteria of selection of a number of factors. These include the domains, expertise seeking factors, contextual factors, etc.

The discussion has been modified to be more specific. Selection criteria for factors have been added. Contextual factors are the factors that have been derived from expertise seeking area. These factors are focuses on human interaction perspective. If we combined these factors to expert finding systems, we could improve their effectiveness.  

At this very general level of discussion, it is unclear who is the targeted reader and what is the value of the study for her/him.

The contribution and the targeted readers of the study have been added in the last paragraph in the introduction section.

Furthermore, some methods, e.g. ones in Tab. 4, could be described in more detail.

Table 4 contains the expertise sources, such as publications, homepages, related documents and social networks; from which we can discover if the person is an expert or not.  The table has been described in more details. Moreover, a brief introduction about expertise sources has been added into introduction section line 86, paragraph 6, to clarify the main concepts of the study for the readers.

Reviewer 2 Report

Focus: The purpose of this study is to analyze existing expert finding systems domains, expertise sources, methods, data sets, the differences between expertise retrieval and seeking and the contextual factors that have been combined into expert finding systems. This study conducted a Systematic Literature Review on the state-of-the-art expert finding systems and expertise seeking studies, published between 2010 and 2019.

Strengths:

Well organized and well written. Formulated questions are answered in detail.

General Comments: A few mistypos are found in text which need to be fixed. The figure captions are not in wrong order.

Abstract: As you mentioned in section three that “[18]” (Kitchenham) approach is applied to conduct this review. Therefore, it is understood that you followed the standerad protocls to design this systematic literature review. So “It applied a review protocol that 20 integrated 2 stages (i.e., manual and automatic) for investigating the published studies” in abstract is no need since it is understood.

Introduction:

Line 54, page 2, “A survey study conducted by [7] analysed expert finding systems techniques, but, it does not cover the methods in a systematic manner and no emphasis was given to the domains that applied expert finding systems…” This is the only study which seems to be related work of your study. Therefore, a short description can not differentiate your work with it. I recommend that authors would consider this and explain in detail by touching a comparison view so that contribution of this study would be appeared clearly.

Research Method: As you mentioned in Figure 2 (The authors mistyped as Figure 1), the result produced by automatic search was 715 publications and manual search got 113 publications. After removal of duplications, authors got 630 studies. After inclusion and exclusion criteria, a huge number of studies were dropped. This gave me a feel that the search string is not much specific. If the string would be more specific, the results would be more specific.

Limitation:

Quality of work and Biasness are two factors mentioned for limitations of this study. Both factors are vague and need more depth explanation of limitations. Three threats to the validity are usually associated with systematic literature review or survey articles such as 1) coverage of relevant studies, 2) biasness on the selection of relevant studies (internal validity), and 3) data extraction inaccuracy. These factors might affect your research study. I would recommend to insert “Threats to the validity” instead of Limitations and explain it in more detail.

Author Response

Hello Dear reviewer 2,

First of all, I want to thank you for your valuable comments. All the comments have been revised. Kindly find the required modifications in the following table and the amended version of the paper attached below. Please do not hesitate to contact me if there is any misunderstanding

Comments

Modifications

Abstract: As you mentioned in section three that “[18]” (Kitchenham) approach is applied to conduct this review. Therefore, it is understood that you followed the standard protocols to design this systematic literature review. So “It applied a review protocol that 20 integrated two stages (i.e., manual and automatic) for investigating the published studies” in the abstract is no need since it is understood.

The sentence has been deleted from the abstract. See the abstract line 8.

Introduction: Line 54, page 2, “A survey study conducted by [7] analysed expert finding systems techniques, but, it does not cover the methods in a systematic manner, and no emphasis was given to the domains that applied expert finding systems…” This is the only study which seems to be related work of your study. Therefore, a short description cannot differentiate your work with it. I recommend that authors would consider this and explain in detail by touching a comparison view so that contribution of this study would be appeared clearly.

The comparison has been explained, started from line 147, the last paragraph.

Research Method: As you mentioned in Figure 2 (The authors mistyped as Figure 1), the result produced by the automatic search was 715 publications, and a manual search got 113 publications. After removal of duplications, authors got 630 studies. After inclusion and exclusion criteria, a huge number of studies were dropped. This gave me a feel that the search string is not much specific. If the string would be more specific, the results would be more specific.

The search string is much specific, but we did not put the string in a double quotation, to reduce the probability of missing related studies. For example, if we search with “expert finding systems” instead of expert finding systems, we will lose these studies (Wang, G.A., et al., ExpertRank: A topic-aware expert finding algorithm for online knowledge communities. Decision Support Systems, 2013. 54(3): p. 1442-1451.

Liu, D., et al., How to Choose Appropriate Experts for Peer Review: An Intelligent Recommendation Method in a Big Data Context. Data Science Journal, 2015. 14.

Hofmann, K., et al., Contextual factors for finding similar experts. Journal of the Association for Information Science and Technology, 2010. 61(5): p. 994-1014. Though, the search with string expert finding systems instead of “expert finding systems” will retrieve every study that contains at least one word of the string, for example, contains expert OR finding OR systems. That is why we found a large number of unrelated studies.

Limitation:

Quality of work and Biasness are two factors mentioned for the limitations of this study. Both factors are vague and need more depth explanation of limitations. Three threats to the validity are usually associated with systematic literature review or survey articles such as 1) coverage of relevant studies, 2) biasness on the selection of relevant studies (internal validity), and 3) data extraction inaccuracy. These factors might affect your research study. I would recommend to insert “Threats to the validity” instead of Limitations and explain it in more detail.

Threats to the validity have been added instead of Limitations, and they explained in more detail. See section 5.

Reviewer 3 Report

First of all, it was a difficult article to read and analyze. It is very long and written in a very intricate way. I do not think at all correct that it should be submitted to a section of Computing and Artificial Intelligence. Anyway, the article needs deep revisions, in my opinion. I will point out some aspects that seem relevant to me.
1. I think the abstract needs to be revised to insert a sentence or two about the conclusions of the review.
2. Some language used seems to me to be weak from the scientific point of view. For example, in line 34 what problems do the authors refer to? In line 32, "people's head" is an inappropriate expression. In line 87, what is meant by "real life"? The core concepts of the study are defined very lightly and without proper reasoning. For example, "expert finding systems" indicates a full page with advantages, but the concept isn't really defined.
3. The citation system is inadequate. The numbering system should be used to locate the citation and not to replace the name of the authors, as in lines 79, 81, 83 or 171. The numbering of the figures is not correct either. There are two Figure 1. Lines 199-201 cite publisher databases and reference databases as all being equivalent, the latter being known to include publications of the former, which generates duplicates and disrupts data collection.
4. The explanation of the methods is very extensive, in my opinion, and it is not feasible for only one of the figures to occupy a full page. I absolutely disagree that there are good and bad journals as it is on line 276. I believe this part should be summed up in essence.
5. The research questions are interesting. And the way the authors answer them is perhaps the best part of the paper. I think the authors should start the paragraphs by the main idea and then argue. Otherwise, it makes reading very confusing and difficult.
6. Future work and conclusions need to be better structured to make it clear what the study's conclusions really are. It is normal for expert finding systems to be absolutely contextual, as they depend on the particularities of each needs area. But the conclusion needs to make clear what is the paper contribution.
7. Finally, I disagree with Appendix A. I believe that the list of references should be one and include everything, referring the numbers to the text and to the tables of the results. No need to create a Study ID list.

Author Response

Hello Dear reviewer 3,

First of all, I want to thank you for your valuable comments. All the comments have been revised. Kindly find the required modifications in the following table and the amended version of the paper attached below. Please do not hesitate to contact me if there is any misunderstanding

Comments

Modifications

1.       I think the abstract needs to be revised to insert a sentence or two about the conclusions of the review.

The abstract has been revised, and sentences about conclusion have been added. See line 7,8,9 and 10 in the abstract.

2.       Some language used seems to me to be weak from the scientific point of view. For example, in line 34 what problems do the authors refer to? In line 32, "people's head" is an inappropriate expression. In line 87, what is meant by "real life"? The core concepts of the study are defined very lightly and without proper reasoning. For example, "expert finding systems" indicates a full page with advantages, but the concept isn't really defined.

Proofreading has been made. For example, problems changed to “problems in a particular domain, such as medical problems”, "people's

  head" changed to “people mind”. The core concepts of the study have been defined in details in the introduction  section.

3.       Include:

a)       The citation system is inadequate. The numbering system should be used to locate the citation and not to replace the name of the authors, as in lines 79, 81, 83 or 171.

b)      The numbering of the figures is not correct either. There are two Figure 1. Lines 199-201 cite publisher databases and reference databases as all being equivalent, the latter being known to include publications of the former, which generates duplicates and disrupts data collection.

a)         The numbering citation system is required by the journal; but the citation such as in lines 79, 81, 83 or 171 has been written in the right way.  You can see this paper (Usability Measures in Mobile-Based Augmented Reality Learning Applications: A Systematic Review) as an example of the required citation system for this journal. After paper’s modification lines 79, 81, 83 became lines 39,41, 42; see paragraph 2 in the Introduction section. 

b)      The numbering of the figures has been modified. The problem of duplication in data collection solved by removing duplicated studies. In Table 2, the duplicated studies have been removed based on the exclusion criteria.

4.       The explanation of the methods is very extensive, in my opinion, and it is not feasible for only one of the figures to occupy a full page. I absolutely disagree that there are good and bad journals as it is on line 276. I believe this part should be summed up in essence.

The explanation of the methods has been reduced, and this section has been summed up in essence.

The figure size has been reduced. The word ‘good’ before journals on line 276 has been removed as you suggested.

5.       The research questions are interesting. And the way the authors answer them is perhaps the best part of the paper. I think the authors should start the paragraphs by the main idea and then argue. Otherwise, it makes reading very confusing and difficult.

Thank you for your comments. The paragraphs have been started with the main idea, as you mentioned.

6.       Future work and conclusions need to be better structured to make it clear what the study's conclusions really are. It is normal for expert finding systems to be absolutely contextual, as they depend on the particularities of each needs area. But the conclusion needs to make clear what the paper contribution is.

Future work has been changed to research findings on identified gaps, and it has been written clearly. The contribution of the study has been clarified in the conclusion section.

7.       Finally, I disagree with Appendix A. I believe that the list of references should be one and include everything, referring the numbers to the text and to the tables of the results. No need to create a Study ID list.

Appendix A has been deleted.

Round 2

Reviewer 1 Report

The Authors have extended the paper according to the remarks of the reviewers.

Reviewer 2 Report

The authors have addressed my concerns. 

Reviewer 3 Report

The authors significantly improved the paper.